Brief Investigation

# Frequent, infinitesimal bottlenecks maximize the rate of microbial adaptation

Oscar Delaney (iD) ,* Andrew D. Letten (iD) , Jan Engelstädter (iD)

School of the Environment, The University of Queensland, Queensland 4072, Australia

*Corresponding author: School of the Environment, The University of Queensland, Queensland 4072, Australia. Email: o.delaney@uq.net.au

Serial passaging is a fundamental technique in experimental evolution. The choice of bottleneck severity and frequency poses a dilemma: longer growth periods allow beneficial mutants to arise and grow over more generations, but simultaneously necessitate more severe bottlenecks with a higher risk of those same mutations being lost. Short growth periods require less severe bottlenecks, but come at the cost of less time between transfers for beneficial mutations to establish. The standard laboratory protocol of 24-h growth cycles with severe bottlenecking has logistical advantages for the experimenter but limited theoretical justification. Here we demonstrate that contrary to standard practice, the rate of adaptive evolution is maximized when bottlenecks are frequent and small, indeed infinitesimally so in the limit of continuous culture. This result derives from revising key assumptions underpinning previous theoretical work, notably changing the metric of optimization from adaptation per serial transfer to per experiment runtime. We also show that adding resource constraints and clonal interference to the model leaves the qualitative results unchanged. Implementing these findings will require liquid-handling robots to perform frequent bottlenecks, or chemostats for continuous culture. Further innovation in and adoption of these technologies has the potential to accelerate the rate of discovery in experimental evolution.

Keywords: population bottlenecks; adaptation rate; beneficial mutations; chemostats; serial passaging; continuous culture; experimental evolution; resource constraints

## Introduction

Experimental evolution is a useful proving ground for both fundamental and applied discoveries in biology, for instance to develop a better understanding of the emergence and spread of antimicrobial resistance (McDonald 2019). The short doubling time of many bacterial species makes them an ideal model system for experimental evolution (Lenski 2017; McDonald 2019). However, rapid growth also means that cells quickly deplete available resources, which in turn curtails growth. Therefore there must be some supply of fresh media and removal of cells. This can either take the form of population bottlenecks—relatively infrequent, usually daily, severe reductions in population size—or, less commonly, frequent small reductions as achieved in a chemostat with a steady outflow rate (Herbert et al. 1956; Gresham and Dunham 2014).

The former approach, known as serial passaging, is considerably more popular and has led to much fruitful work, most notably the long-term evolution experiment evolving *Escherichia coli* for more than 60,000 generations (Lenski 2017). While efficient and high throughput, serial passaging may be suboptimal when the goal is to maximize the rate of evolution: the very beneficial mutations required for adaptive evolution to occur are frequently lost to population bottlenecks of 100:1 or greater severity (Wahl et al. 2002).

There has been a rich vein of theoretical study over the last two decades into the relationship between the severity and frequency of population bottlenecks and the fraction of beneficial mutations that are lost (Wahl and Gerrish 2001; Heffernan and Wahl 2002;

Wahl et al. 2002; Hubbarde et al. 2007; Hubbarde and Wahl 2008; Patwa and Wahl 2008; Campos and Wahl 2009, 2010; Wahl and Zhu 2015; Bittihn et al. 2017; LeClair and Wahl 2018). While a number of studies have indicated that bottlenecks of intermediate size (e.g. ~8:1) are optimal (Wahl et al. 2002; Hubbarde and Wahl 2008), here we show that revised assumptions yield markedly different conclusions. Our analysis corroborates that the severe dilution ratios (100:1 or 1,000:1) routinely employed in experimental evolution are indeed suboptimal. However, we find that rates of evolution are in fact optimized under infinitesimal bottlenecks infinitesimally close together, as is the case in a classical chemostat system. If logistical constraints require a 24-h growth period, we show that even then a less severe bottleneck of ~5:1 maximizes the adaptation rate.

## Analytical predictions for exponential growth

Resource-constrained growth is more realistic, and we address this in a later section, but first we consider the simpler case of unconstrained growth between bottlenecks. Following Wahl et al. (2002), we consider a clonal population of bacteria that grows exponentially for $\tau$ time units at a growth rate of $r$. At time $t = \tau$ the population attains its maximum size of $N$ cells and a bottleneck is applied where each cell has a $D \in (0, 1)$ chance of surviving and entering the next growth phase. To maintain a stable population over time, we set $D = e^{-r\tau}$ such that the population after the bottleneck returns to its

initial size of $DN$. A beneficial mutation occurs with frequency $\mu$ at each cell division, and the selective benefit of the mutation is drawn according to $s \sim \text{Exp}(\frac{1}{\omega})$, yielding a growth rate for the mutant of $r(1+s)$. We seek the value of $D$ (or, equivalently, $\tau$) that will maximize the rate at which beneficial mutations reach fixation.

Consider a mutation of selective benefit $s$ that arises at time $t$. We model the growth of this mutant as a pure-birth process where the time taken for each cell to divide is identically and independently distributed according to $\text{Exp}(r(1+s))$. Such processes have been extensively mathematically analyzed, and it has been known since Yule (1925) that at time $u$ the number of individuals $X(u)$ in a pure-birth process with $X(0)=1$ and a growth rate of $\lambda$ is distributed according to $X(u) \sim \text{Geom}(e^{-\lambda u})$. Let us denote by $M(\tau^-)$ and $M(\tau^+)$ the mutant population sizes immediately before and after the bottleneck, respectively. Then, the number of mutants at the end of the growth phase is distributed according to $M(\tau^-) \sim \text{Geom}(\beta)$ with $\beta = e^{-r(1+s)(\tau - t)}$. The population then undergoes a bottleneck, where each cell has survival probability $D$, and thus the number of mutants after the bottleneck is distributed according to $M(\tau^+) \sim \text{Bin}(M(\tau^-), D)$.

We can now apply the law of total probability to find the probability $P_i$ that there will be exactly $i$ mutants immediately after the first bottleneck:

$$
\begin{aligned}
P_i &= P(M(\tau^+) = i) \\
&= \sum_{j=1}^{\infty} P(M(\tau^-) = j)P(M(\tau^+) = i \mid M(\tau^-) = j) \quad (1) \\
&= \sum_{j=1}^{\infty} \left( \beta(1-\beta)^{j-1} \right) \left( \binom{j}{i} D^i (1-D)^{j-i} \right) \quad (2)
\end{aligned}
$$

Further, if exactly $i$ mutants are present immediately after the bottleneck, then each of these will have separate and independent evolutionary trajectories, provided that $N \gg i$. In particular, the probability that there will be no descendants in the distant future is

$$
V(t, s) = \sum_{i=0}^{\infty} P_i V(0, s)^i \quad (3)
$$

This equation can first be solved in the simpler case where $t = 0$, which yields

$$
V(0, s) = \frac{D^{-1} - 1}{D^{-(1+s)} - 1} \quad (4)
$$

Substituting equation (4) back into equation (3) and solving returns the general solution for any $t \in [0, \tau)$:

$$
V(t, s) = \left( 1 + \frac{1 - D^s}{\beta(D^{-1} - 1)} \right)^{-1} \quad (5)
$$

The rate of beneficial mutations occurring that will go on to fix can now be determined by integrating over all times in the growth period and all possible selective benefits:

$$
\gamma = \frac{1}{\tau} \int_0^{\infty} \overbrace{\frac{1}{\omega} e^{-\frac{1}{\omega}s}}^{\substack{\text{mutation has} \\ \text{selective benefit } s}} \int_0^{\tau} \overbrace{\mu ND re^{rt}}^{\substack{\text{mutation rate} \\ \text{at time } t}} \overbrace{(1 - V(t, s))}^{\substack{\text{mutation} \\ \text{survives}}} dt\, ds
$$
$$
\approx r\mu N \cdot \frac{\omega}{1 + \omega} \cdot \frac{\ln(D^{-1})}{D^{-1} - 1} \quad (6)
$$

There appears to be no closed-form solution to this double-integral; however, the approximation provided is an excellent fit for all $D$ and for all reasonable selective benefits ($\omega \lesssim 1$). We refer the interested reader to our algebraic manipulations implemented

in Mathematica v13.1.0.0 (Wolfram Research Inc 2022), including exact expressions where applicable (see *Data availability*).

Several features of the approximation in equation (6) are salient. Unsurprisingly, the rate of beneficial mutations fixing is proportional to the growth rate, the mutation rate, the population size, and the average selective benefit $\omega$ (at least for small values of $\omega$). The relationship with $D$ is nontrivial but can be more simply approximated by $\gamma \propto \sqrt{D}$ when D is not too small ($D \gtrsim 0.1$). The key observation is that $\gamma$ is monotonically increasing in $D$ and thus $D \to 1$ maximizes the adaptation rate.

We can also now approximate the adaptation effective population size, which is the size of an ideal Wright–Fisher population that would evolve at the same rate as the fluctuating population in question (Campos and Wahl 2009). Following the method in Campos and Wahl (2009) of letting $\gamma = N_e \mu\, 2\, \omega$ and solving for $N_e$ we find that:

$$
N_e \approx N \frac{r \ln(D^{-1})}{2(D^{-1} - 1)} \quad (7)
$$

This differs from Campos and Wahl (2009) by a factor of $\frac{1}{2(1-D)}$ such that the adaptation effective population size goes to $N$ rather than 0 as $D \to 1$, which conforms better with expectations.

## Numerical simulations

To better understand the underlying dynamics, and to corroborate our theoretical solution, we created a computer simulation of the theoretically analyzed scenario in the programming language R v4.3.0 (R Core Team 2023). A population of clonal bacteria was initialized, and the Gillespie algorithm was used to model random mutation and cell division events (Gillespie 1976; Johnson 2019). Bottlenecks were implemented by drawing random numbers from a binomial distribution with a probability $D$ of each cell surviving. Figure 1 shows example population dynamics for a single short run of the simulation.

Figure 2 shows the adaptation rate for a range of commonly used values of $D$. Our theoretical results match the simulations closely, corroborating each other.

We found similar results to Wahl *et al.* (2002) that the average selective benefit of mutations that eventually fix is $2\omega$, which is double that of all mutations that occur (Supplementary Fig. 1). Also following Wahl *et al.* (2002), we found that the occurrence time of ultimately successful mutations follows an approximately uniform distribution (Supplementary Fig. 2). More mutations occur later in a growth phase when the population is larger, but these mutants have less time to multiply, so they are more prone to being lost at the first bottleneck, and these two competing effects roughly cancel out.

## Resource-explicit growth

Thus far, we have assumed growth is exponential, which is unrealistic. We employ a mechanistic resource-explicit model, with the growth rate determined by a Monod function $\frac{rR}{k+R}$, where $R$ is the current concentration of a single growth-limiting resource and $k$ is the resource concentration when the growth rate is half its maximum value (Monod 1949; Wahl *et al.* 2002). Each cell division is modeled as consuming one unit of resource. Here, at each bottleneck, the population is diluted by a factor of $D$ into fresh media with resource concentration $R_0$. The resulting population dynamics are similar to those of the resource-unconstrained model (Supplementary Fig. 3). We also created a true continuous

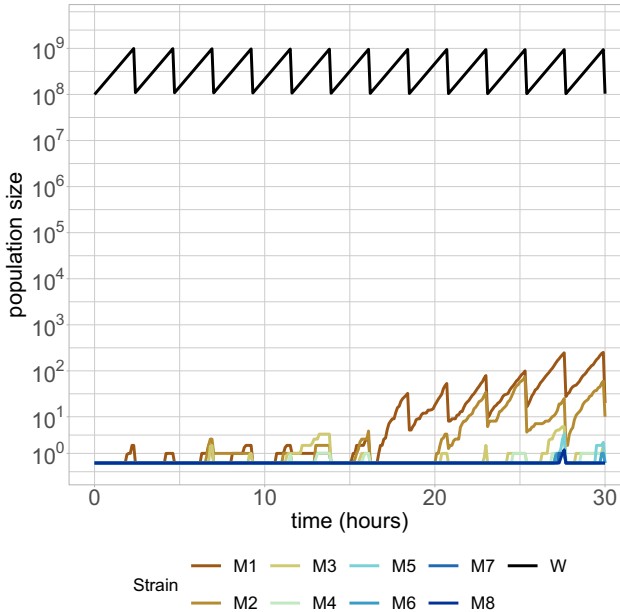

**Fig. 1.** One run of the simulation for the resource-unconstrained model. Parameters used were $N = 10^9$, $\mu = 3 \times 10^{-9}$, $r = 1$, $\omega = 0.1$, $D = 0.1$, $\tau = \ln(10)$.

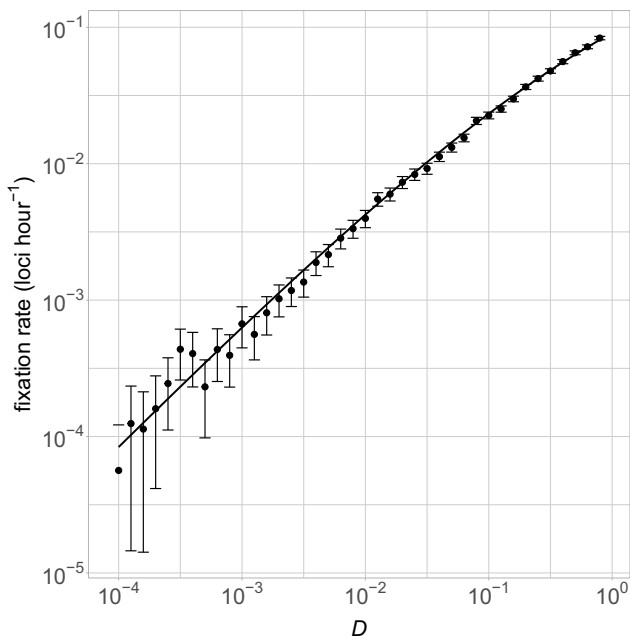

**Fig. 2.** Optimal bottleneck size in resource-unconstrained growth. Each dot represents 1,000 simulation runs for 50 simulated hours each. The rate of beneficial mutations fixing was calculated for each run, and error bars were calculated as the mean $\pm 1.96 \times$ se. The solid line shows the theoretical solution in equation (6). Parameter values used were $N = 10^9$, $\mu = 10^{-9}$, $r = 1$, $\omega = 0.1$, $\tau = -\ln(D)$.

culture simulation where there were no bottlenecks, but a constant rate of fresh media supplied and reaction volume removed.

We extended the analytical solution for resource-unconstrained growth to suit the resource-explicit model. Since the growth rate varies over time, instead of the maximum growth rate $r$ the appropriate input to equation (6) is the average growth rate which is given by $\bar{r} = \tau^{-1} \ln(D^{-1})$. Moreover, the maximum wild-type population size $N$ may now vary between growth periods. Instead, we used the value of $N$ once an equilibrium between resources and population has been reached, $N^*$. Thus, the final equation for the adaptation rate general to resource-constrained or unconstrained growth is:

$$\gamma \approx \frac{1}{\tau} \cdot \mu N^* \cdot \frac{\omega}{1 + \omega} \cdot \frac{(\ln D)^2}{D^{-1} - 1} \qquad (8)$$

For the resource-constrained simulations, we let $\tau$ and $D$ vary independently, and optimized $\gamma$ over both degrees of freedom. Figure 3 shows that here too the adaptation rate is maximized with large $D$ and small $\tau$. Supplementary Fig. 4 shows that the simulation and analytical results also match very well for the resource-constrained scenario, providing some evidence that both have been implemented appropriately. Moreover, for a variety of values of $k$, $D \to 1$ continues to optimize the adaptation rate (Supplementary Fig. 5). Supplementary Fig. 5 also uses the true chemostat simulation model for $D = 1$, and these results are a smooth continuation from $D < 1$ chemostat approximations with short growth periods and small bottlenecks. This suggests that as expected, frequent bottlenecking approaches a true chemostat model in the limit as $D \to 1$.

Of particular interest to experimentalists is the special case where $\tau = 24$ h as that is the most convenient period with which to perform serial passaging experiments and so has become standard practice. As seen in the top row of grid squares in Fig. 3, and also shown in more detail in Supplementary Fig. 6, the optimal value of $D$ is now approximately 0.2, which gives

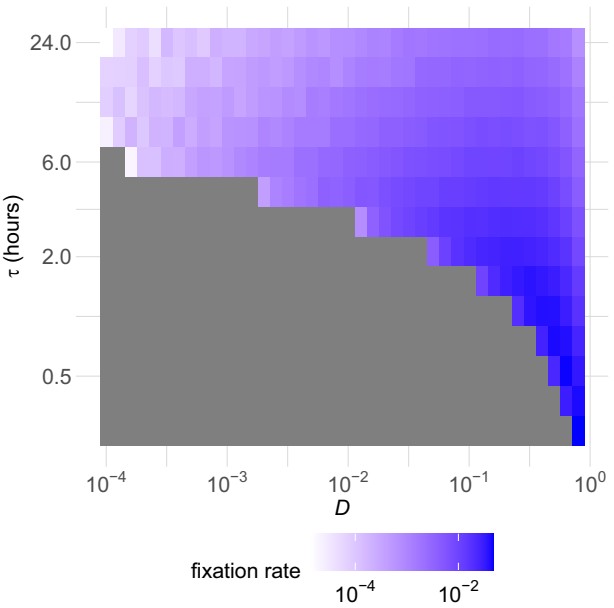

**Fig. 3.** Adaptation rate in the $D - \tau$ landscape. Each grid square represents 1,000 simulation runs. Parameters are the same as in Fig. 2 except for $\tau$ varying and $R_0 = k = 10^9$, $r = 1.5(1 + \frac{k}{R_0}) = 3$. The maximum growth rate is higher than in the resource-unconstrained model to prevent the population from going extinct.

adaptation rates an order of magnitude higher than the commonly used value of $D = 0.001$.

Finally, we created a separate genotype-based simulation model, where mutations occurred at one of a fixed small number of loci, and multiple segregating mutations could be present in the same genome. In this genome-based model, clonal interference

caused the adaptation rates to be slower (results not shown), but qualitatively the same finding emerged that $D \to 1$ is optimal (Supplementary Fig. 7).

## Discussion

Contrary to standard experimental practice, we found that the rate of adaptive evolution is maximized when bottlenecks are frequent and small. To understand why our results are markedly different from previous theoretical studies, it is useful to consider each of the three revised assumptions in our method: time optimization rather than transfer optimization; binomial sampling rather than a Poisson approximation; and stochastic rather than deterministic growth. Notably, implementing any one of these three changes alone is insufficient to generate our main finding that continuous culture is optimal for maximizing rates of evolution.

*Time optimization:* We sought to maximize the adaptation rate per unit time. This contrasts with the literature, which since the original exploration in Wahl *et al.* (2002) has instead usually maximized the adaptation rate per transfer. If the goal is to complete an experiment quickly, and thus the limiting factor is calendar time, then our approach makes sense. However, in some circumstances, the calendar time may matter less than the experimenter's time, and since the experimenter is only required once per growth period at the transfer step, maximizing the adaptation rate per transfer may best conserve the experimenter's time. Indeed, if the experimenter is required to perform every transfer, our result of very frequent transfers optimizing the rate of adaptation would be an irrelevant curiosity. Chemostats provide one solution to this conundrum by removing the manual dilution step. Alternately, liquid-handling robots are increasingly able to run experiments independently (Tegally *et al.* 2020). Either of these solutions may in fact save experimenter time by removing the need for humans to perform serial transfers at all. This technological progress may make optimization per unit of time more relevant than in the past.

*Binomial sampling:* The original literature assumed that the population after the bottleneck is drawn with replacement from the gene pool (Wahl and Gerrish 2001; Wahl *et al.* 2002), which entails $M(\tau^{+}) \sim \mathrm{Bin}(DN, \frac{M(\tau^{-})}{N}) \approx \mathrm{Poi}(DM(\tau^{-}))$. However, this is problematic as it allows the number of mutants after the bottleneck to exceed the number of mutants before the bottleneck, which is impossible. Instead of randomly drawing cells with replacement until we have the correct number, the appropriate approach introduced first in Hubbarde *et al.* (2007) is to model whether each cell survives the bottleneck as independent Bernoulli random variables with parameter $D$ (Hubbarde and Wahl 2008; Wahl and Zhu 2015). Summing these random variables over the number of mutants present before the bottleneck, sampled without replacement, yields the formula we employ: $M(\tau^{+}) \sim \mathrm{Bin}(M(\tau^{-}), D)$. The Poisson approximation is no longer suitable with this binomial distribution as $D$ may be near 1 and the approximation only works with a probability near 0. Replacement sampling is steeped in the standard population genetic paradigm, where populations are infinite and individuals are drawn with alleles randomly chosen from this infinite gene pool. While this is often a useful approximation to reality, in this case, the small population size of the mutant lineages—by definition initially just one cell—means that the infinite approximation fails completely.

*Stochastic growth:* Some early work used discrete generation times (Heffernan and Wahl 2002) or modeled the population as growing exactly exponentially over time (Wahl and Gerrish 2001). We followed later approaches by modeling a population with overlapping generations where the times until division for each cell are drawn from independent exponential distributions (Hubbarde *et al.* 2007; Hubbarde and Wahl 2008). These stochastic effects could be relevant given the initially small mutant population sizes (Patwa and Wahl 2008).

Somewhat surprisingly, it is only through the combined use of both binomial sampling and time optimization that we observe our main result (Fig. 4). Supplementary Fig. 8 shows more mechanistically how changing from a Poisson approximation to the full binomial distribution makes a large difference, but the deterministic growth approximation works quite well. It is straightforward that the switch to time optimization makes a large difference, as the adaptation rate $\gamma$ differs by a factor of $\tau$.

Using these revised assumptions, we have shown that evolution experiments could yield 10–100 times faster adaptation by switching from serial passaging to continuous culture. This is a surprising result, as prima facie it suggests experiments that have previously taken decades could be performed in months. One possible explanation is that the sheer magnitude of the superiority of continuous culture over serial passaging for accelerating rates of evolution has not previously been explicated. Another explanation, perhaps more likely, is that chemostats have significant practical limitations, including contamination risks, more careful and error-prone setup requirements, and being less parallelizable than batch culture. This may cause experimentalists to continue preferring batch culture despite the theoretical appeal of continuous culture.

Regarding the practical challenges of chemostats, as more experimentalists come to favor continuous culture, innovations in

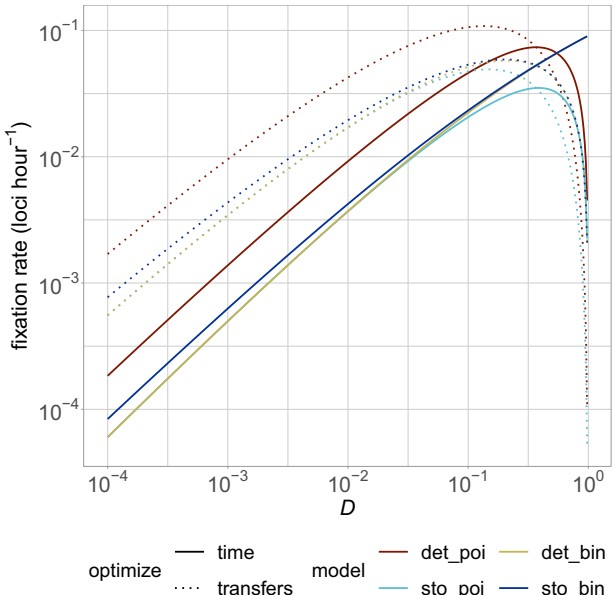

**Fig. 4.** Resource-unconstrained analytical results for different assumptions. Our model was time-optimized, stochastic, and binomial. For comparison, the original analysis in Wahl *et al.* (2002) was transfer-optimized, deterministic, and Poisson approximated, while Hubbarde and Wahl (2008) and Wahl and Zhu (2015) updated this to use the exact binomial distribution and stochastic growth. The formulas for each of these eight lines are available in the R code. In all cases, $N = 10^{9}$, $\mu = 10^{-9}$, $r = 1$, $\omega = 0.1$, $\tau = -\ln(D)$.

experimental design and technology could proliferate, eventually overcoming the current difficulties. Indeed, a number of open-source chemostat designs have been described in recent years, with several biotech startups offering prebuilt arrays at a significantly reduced cost compared to better-known commercial platforms (Miller *et al.* 2013; Wong *et al.* 2018; Ekkers *et al.* 2020; Steel *et al.* 2020).

The original pair of papers on which we are building (Wahl and Gerrish 2001; Wahl *et al.* 2002) received considerable attention, with later work extending the analysis to more complex settings. Two closely related articles considered a burst–death life-history model where each cell generates some (perhaps >1) number of offspring at a constant burst rate, and dies at a constant death rate (Hubbarde *et al.* 2007; Hubbarde and Wahl 2008). This extension of the model did not qualitatively change the result, with $D \approx 0.2$ still being optimal. Hubbarde and Wahl (2008) did implement binomial sampling and stochastic growth, and while their mathematical working was very different to ours, relying on probability generating functions, their final approximation in equation (12) is the same as our equation (8) except for dividing by $\tau$ and a factor of $1 + \omega$, which disappears for small-effect mutations. We incorporated the continuous-time nature of this model, but have not replicated their approach of allowing more than one simultaneous birth and death between bottlenecks. This could be the subject of future exploration.

Campos and Wahl (2009, 2010) extended this work to also consider the effect of deleterious mutations and clonal interference on adaptation rates, but focused on estimating the severity of clonal interference and the adaptation effective population size, rather than optimizing $D$. Clonal interference, the competition between different adaptive lineages vying for fixation in asexual populations, is a key feature of bacterial adaptive evolution (Park and Krug 2007) and deleterious mutations are the norm not the exception so are important to consider. Campos and Wahl (2009, 2010) relied on the Poisson approximation and deterministic growth assumptions of Wahl and Gerrish (2001) to estimate the adaptation effective population size, and because we revised these assumptions, our approximation in equation (7) is considerably different. Our result has the more reasonable property that the effective population size does not approach zero in the continuous culture limit where population size is constant (and nonzero). We also revisited the clonal interference modeling, and found that including this leaves the basic outlook unchanged, with continuous culture being optimal (Supplementary Fig. 7).

The original model has also been extended to consider different types of beneficial mutations, such as those that shorten generation times, delay the onset of stationary phase, and lower mortality (Wahl and Zhu 2015). This extension reaffirmed the original finding in Wahl *et al.* (2002)—which differs from ours—that the optimal dilution ratio is $D \approx 0.2$ across a wide range of parameter settings and types of beneficial mutations. Adapting the modeling in Wahl and Zhu (2015) to the altered assumptions in this work provides another avenue for future research.

In an earlier contribution investigating the cognate problem of minimizing mutation rates in synthetic biology, Bittihn *et al.* (2017) raised several points that overlap with those described here. Although they use a mathematically more complex approach based on diffusion approximations, Bittihn *et al.* (2017) also challenge the assumption that optimizing for adaptation per transfer is best, and question the approach to binomial sampling in Wahl and Gerrish (2001). They instead used a hypergeometric distribution where at each bottleneck exactly $N_0$ individuals survive, of which a variable number are mutants. We nevertheless maintain that it is more realistic to treat the survival of each cell as a set of independent random variables, rather than constraining the total number of survivors to be constant.

Despite the wealth of theoretical literature on this topic, there has been very little empirical investigation of the relationship between bottleneck size and adaptation rate. To our knowledge, the only such study is Chavhan *et al.* (2019), where the authors serially passaged *E. coli* for 380 generations under different bottleneck severities. They found that all else equal, with a fixed value of $\tau = 24$ h, a bottleneck size of $D = 10^{-1}$ led to faster adaptation than under very severe bottlenecks of $D = 10^{-6}$. While this finding is interesting, and directionally supports our conclusion that smaller bottlenecks are better, much more experimental work is needed here, particularly to try more different values of $D$, and to vary $\tau$ as well.

We have shown that continuous culture should lead to rates of adaptive evolution 10–100 times higher than commonly seen in serial passaging experiments. Notwithstanding the aforementioned constraints on throughput, we venture that greater uptake of methods for continuous culture (e.g. chemostats) would be of significant benefit to research in experimental evolution. A faster pace of experimental evolution can accelerate the generation of new scientific knowledge, which is valuable both for its own sake and because of the many applications of evolutionary biology to societally important domains, including cellular agriculture and antimicrobial resistance research.

## Data availability

Additional figures are available in the supplementary file. All R code used for the simulation and generating the figures, as well as the Mathematica notebook used to perform algebraic manipulations, are available at https://github.com/Oscar-Delaney/optimizing-bottlenecks. Supplementary material is available at GENETICS online.

## Acknowledgments

We thank James Richardson for useful discussions about the mathematical argumentation in this paper, Lindi Wahl for helping contextualize our contribution in the broader literature, and Yashraj Chavhan for linking our theoretical work to existing experimental findings.

## Funding

OD receives a Vice-Chancellor's Scholarship at the University of Queensland. ADL is supported by Australian Research Council grants DP220103350 and DE230100373. JE is supported by Australian Research Council grant DP190103039.

## Conflicts of interest

The author(s) declare no conflict of interest.

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

*Editor: A. Agrawal*