## [Peer Review File · Genetics]

Frequent, infinitesimal bottlenecks maximize the rate of microbial adaptation

Oscar Delaney, Andrew Letten, and Jan Engelstaedter

NOTE: The reviews and decision letters are unedited and appear as submitted by the reviewers.

In extremely rare instances and as determined by a Senior Editor or the EIC, portions of a review may be redacted. If a review is signed, the reviewer has agreed to no longer remain anonymous.

The review history appears in chronological order.

Review Timeline:

Submission Date:	2023-08-09
Editorial Decision:	2023-09-18
Resubmission Received:	2023-09-26
Accepted:	2023-10-02

September 17, 2023

GENETICS-2023-306417

Frequent, infinitesimal bottlenecks maximize the rate of microbial adaptation

Dear Dr. Delaney:

First, I apologize for the long delay. I waited for many weeks for a reviewer who never delivered. I have decided to proceed based on my own reading of the manuscript as well as the high level of expertise of one reviewer, who identified herself as Prof. Lindi Wahl. I am pleased to inform you that, with minor revisions, it is potentially suitable for publication in GENETICS.

As you know, Prof. Wahl is intimately familiar with this literature and has been a major contributor to it. She was generally positive about your work, though had some significant concerns. These should all be relatively easy to address in revision but I urge you to take them seriously.

Her first major concern is about the way the story is presented and I recommend you follow her suggestion in regards to different choices of what is being optimized. I found her comments about robots vs. manual transfers interesting and relevant in this context.

Several of her other major comments point to misrepresentations of what is in the literature. This needs to be corrected. I don't know this literature myself but have every reason to trust she is correct. If you disagree, you will need to make a strong and explicit case that I can have assessed. Prof. Wahl's review is positive in its tone and aims to help you produce a manuscript that accurately reflects the literature and will be as informative as possible for your future readers; I hope you take her comments in that light.

I have only one comment of my own. What is the reason for the discrepancy between your equation 7 and the result of Campos and Wahl (2009) that you mention on line 58 of p. 2?

We look forward to receiving your revised manuscript. Please let the editorial office know approximately how long you expect to need for revisions.

Upon resubmission, please include:

1. A clean version of your manuscript;
2. A marked version of your manuscript in which you highlight significant revisions carried out in response to the major points raised by the editor/reviewers (track changes is acceptable if preferred);
3. A detailed response to the editor's/reviewers' comments and to the concerns listed above. Please reference line numbers in this response to aid the editors.

Additionally, please ensure that your resubmission is formatted for GENETICS.

<https://academic.oup.com/genetics/pages/general-instructions>

Follow this link to submit the revised manuscript: Link Not Available

Sincerely,

Aneil Agrawal
Associate Editor
GENETICS

Approved by:
Nicholas Barton
Senior Editor
GENETICS

Reviewer #1 (Comments for the Authors (Required)):

Overall this is a nice contribution that addresses a topic that has been generating some interest recently, a couple decades after

the initial idea. I have no issues at all with the science, the conclusions, or the support for those conclusions... everything looks great on that front. My only comments relate to putting the work correctly in the context of previous contributions. I have signed my review since I think it will be impossible to retain anonymity here, and this will be easier for everyone in this case!

For context, I feel I must point out that when this idea first came up, we considered optimizing adaptation rates *per unit time*, rather than per transfer. As the authors mention, we found those results were trivial (no bottlenecks, that is, constant population size = continuous culture is best) and, more importantly, that these results were irrelevant to the experimental community ("an irrelevant curiosity", as the authors write in this Discussion, summarizes our thinking perfectly!).

At that time of course every transfer was done laboriously by hand (most still are, I think (?) since liquid handling robots are still pretty expensive and experimentalists tell me they are still a little "drippy and spilly"). I thought we had mentioned this decision (to optimize per transfer) in one of those early papers, but I can't find it and so I don't blame you for not finding it either. I certainly mentioned it many many times during talks I gave about this work... that we were intentionally optimizing adaptation rate per transfer, because transfers were what mattered experimentally. (We also intentionally optimize adaptation for a given final population size, and optimize total flux rather than just fixation probability, aspects discussed in LeClair and Wahl.)

Major Comment 1: I guess I'm asking you to take my word for it, but I would suggest changing the "story" of your contribution to something more along the lines of: "earlier work optimized adaptation rates per transfer, intentionally and for good reasons. Now we have liquid handling robots and so optimizing per unit time might be more than an irrelevant curiosity, if total experimental calendar time is more important than the number of transfers that have to be accomplished. So we decided to revisit this question with a different metric of optimization." This would be a lot more fair to our previous work than making the argument that the previous assumptions were flawed. Later, in the Discussion, you point to three "improvements" over previous work. Again, I would argue that changing the metric is not an improvement, per se, but a different choice of what to optimize; you're answering a different question (which is great!).

Major comment 2: Along the same lines, I think figure S8 would be interesting in the main text. I would strongly suggest deleting the Poisson lines (we've known for a long time this is quite a "wrong" approximation, see next point, so that's a bit of a straw man argument), but showing the other cases to highlight how we get different answers depending on whether we optimize per transfer or per unit clock time is interesting. Time optimized versus transfer optimized? I also think it is interesting to show how deterministic versus stochastic growth models differ, and by how much, since that hasn't really been addressed before. Showing how wrong the wrong model is (the Poisson) isn't particularly useful and just adds a lot of clutter.

Major comment 3: Binomial sampling. It's simply not true that "the existing literature since Wahl and Gerrish 2001 generally assumes that the popn ... is drawn with replacement". With the exception of those first couple of papers, ALL of the work on this topic was done using Bernoulli sampling of the mutant lineage. You can find that in Hubbarde 2007, Hubbarde 2008, Patwa 2008, Patwa 2010, Wahl and Zhu 2015, etc. In other words, the field definitely moved on from those early approximations and used exactly the same Bernoulli sampling, yielding a binomial distribution, that you employ here. Since it's pretty well-understood now (for about 15 or more years) that the Poisson approximation wasn't the right way to go, I don't think it deserves much focus in a 2023 paper. So the simple fix for your contribution is to say right from the outset that while the first two papers used a Poisson approximation, later work used the Bernoulli and you're of course following the later work here.

Major comment 4: Stochastic growth: Likewise, once we got our head around how to correctly handle the probability generating functions in continuous time, almost all of our work after around 2001 or 2002 used stochastic growth models. Again, you'll see stochastic growth models with overlapping generations in all of the following papers: Hubbarde 2007, Hubbarde 2008, Patwa 2008, Patwa 2010, Wahl and Zhu 2015, etc. So again you can simply say that you're following the later work in your approach here.

Major comment 5: Resource-constraints. Wahl et al 2002 did not use a phenomenological resource constraint; we used a resource-explicit model, with the same Monod growth rate function you use here (see $\Psi(R)$ in equation 2 in that paper). We also found it made very little difference to anything. So again you can simply make the argument that you're asking a different question (what happens if we optimize per unit time) using the same model that has been studied before.

Overall: Clearly there's going to be some bias in my review, but I honestly think that you're asking a really interesting, different question and that that, in and of itself, makes for a great contribution to the literature. I don't think you need to make arguments about "improved theoretical foundations", given that (1) the choice to optimize per bottleneck was a conscious decision on our part for the 2001 paper, and technical improvements now make a different choice interesting and (2) the field has been using Bernoulli sampling and stochastic, continuous time growth models with overlapping generations for about 15 years now!

I also have to say thank you for reading so many of my ancient papers and I take full responsibility for all the typos, omissions and bad approaches therein.

Signed,
Lindi Wahl

Associate Editor Comments:

Oscar Delaney
Phone +61 422 465 318
o.delaney@uq.net.au

Goddard Building
School of the Environment
University of Queensland
Brisbane 4072
Queensland, Australia

September 26, 2023

Dear Editors,

We would like to thank you and Lindi Wahl for the helpful and generous engagement with our manuscript entitled “Frequent, infinitesimal bottlenecks maximize the rate of microbial adaptation”.

Here we address all the comments raised and outline how we have made appropriate updates to the manuscript (see attached marked-up and clean versions). First we address Lindi Wahl’s Major comments 1 through 5:

1. Thank you for this useful reframing of our contribution. We have made changes accordingly, removing references to ‘improvements’ (p4, line 7; p5, line 99) and explicitly referring to how technological changes lead to time-optimization becoming more interesting to explore (p4, lines 28-30). We also edited the abstract (p1, lines 8-10) to emphasize the time-optimization change and remove discussing the Poisson approximation as we no longer wish to foreground that change. We also removed the phrase ‘questionable assumptions’ from the introduction, because while true that the assumptions can be and indeed are being questioned, it gives an unfairly negative tone; instead we ‘revise’ the assumptions (p1, lines 34-36).
2. We agree that ‘transfer-optimized’ is a better term to describe the earlier literature, so we have employed this new wording throughout the text and in the relevant figure (p4 lines 4, 19, p5 line 107). As suggested we have moved what used to be Figure S8 into the main text (now Figure 4). We also agree that it is inaccurate to imply that much of the past work was using the Poisson approximation, so have updated the caption of Figure 4 to specifically note which model we use, which model the original Wahl 2002 paper used, and which model the later Wahl work used. However, we think that the Poisson approximation discussion and lines in the figure should not be removed altogether as even though that model has not been endorsed for some time, it provides a useful historical perspective to contextualise how this modelling has evolved (see further discussion in major comment 3). In terms of adding clutter, we think the plot remains interpretable with all eight lines, and that adding this detail, while not interesting to all readers, provides some value at little cost.
3. We agree that some of the later literature, such as Hubbarde (2007, 2008) and Patwa (2008, 2010), models Bernoulli trials appropriately. We have added a sentence more properly acknowledging the contribution of Hubbarde 2008 (p5, lines 57-63). However, not all the later literature implements bottlenecks optimally, for instance Campos and Wahl (2009, 2010) approximate the fixation probability of a mutant arising at time t as $2rs_b\tau e^{-rt}$. This approximation is from the original Wahl 2001 paper, and is cited uncritically in the 2009 and 2010 papers (see quotation below). We do not doubt that Dr. Wahl and others were well aware of the problematic Poisson approximation, however this has never been made explicit in the literature. As Dr. Wahl points out in her review, we are not the first to implement binomial sampling correctly, and we have updated the relevant paragraph (p4, lines 31-32, 40) to reflect this. We think it is useful to retain our discussion of why

the original methodology was flawed, so that future readers do not need to deduce for themselves why the methodology changed between Wahl 2001 and Hubbarde 2007. In summary, we have followed the advice in major comment 3 to acknowledge that not all the past models were wrong in this regard, but we have not removed the paragraph because we think the explanation would benefit a reader less acquainted with this history.

4. We thank the reviewer for raising this, and agree our representation of the literature was inaccurate. We have updated the paragraph on Stochastic growth (p5, lines 2-15) to reflect this.
5. We have deleted the sentence in question (p3, lines 3-4) as requested. We were aware that the paper in question (Wahl 2002) originally described Monod resource-constrained growth in equation 2, however later on page 967 it says “Unfortunately the model described in system 2 is somewhat unwieldy for our purposes, and so in this section we have chosen to use a simpler model of population growth” which in equation 6 is phenomenological. So a resource-explicit model was introduced, but then for some of the methodology a simpler phenomenological model was used. This nuance isn’t that important though, so removing the sentence seemed the best solution.

In regards to the editor’s comment re equation 7, we agree that this was under-described. It would be difficult to unpack this further in the analytical predictions section without some of the information in the Discussion, so instead we have expanded our consideration of Campos and Wahl (2009, 2010) to comment on our different adaptation effective population sizes (p5, lines 75-88). This relates to major comment 3 above, as while Campos and Wahl (2009) don’t explicitly use the flawed Poisson approximation, they do rely on a key equation from earlier work, see this quote (Campos and Wahl 2009, p.951):

Following Wahl and Gerrish (2001), we use a continuous time approximation to the Wright–Fisher model described above. Assuming that the wild-type population grows exponentially at rate r during the growth phase, the probability that a new beneficial mutation ultimately survives bottlenecks can be approximated as $2r s_b \tau e^{-rt}$, where t is the time, during the growth phase, at which the new mutation first arises.

Unrelated to the official review process, the other update we made (apart from a few minor wording changes) was to include a paragraph (p5 line 115 - p6 line 3) discussing experimental investigations of the topic, thanks to the lead author of a relevant paper contacting us after reading our preprint.

We hope our revised manuscript addresses any concerns raised by the reviewer and yourself.

Sincerely,

Oscar Delaney
(on behalf of all authors)

October 1, 2023

RE: GENETICS-2023-306503

Mr. Oscar Delaney
The University of Queensland
School of the Environment
Goddard Building (08)
The University of Queensland
Saint Lucia 4072
Australia

Dear Dr. Delaney:

Congratulations! We are delighted to inform you that your manuscript entitled "Frequent, infinitesimal bottlenecks maximize the rate of microbial adaptation" is acceptable for publication in GENETICS. Many thanks for submitting your research to the journal.

Thanks for attending to the revisions. I have one minor suggestion that you can choose (or not) to implement. In relation to eq. 7, you explicitly state that your N_e does NOT go to zero as $D \rightarrow 1$ (unlike the Campos & Wahl result) but I think it is worth explicitly pointing out what your N_e does do in that limit, i.e., $N_e \rightarrow N$, as one would expect, right?

To Proceed to Production:

1. Format your article according to GENETICS style, as discussed at <https://academic.oup.com/genetics/pages/general-instructions>, and upload your final files at <https://genetics.msubmit.net>. Staff will review your files and be in touch if there are any questions.
2. After staff check your files, they will be transmitted to OUP for processing. You will then receive an email with a link to sign your license to publish. Please add jnls.author.support@oup.com and genetics.oup@novatechset.com (or the domains @oup.com and novatechset.com) to your email program's "safe senders" list. Publication cannot occur till the license is signed. Invoices are generated after the license is signed.
3. Your manuscript will be published as-is (unedited-as submitted, reviewed, and accepted) at the GENETICS website as an Advanced Access article and deposited into PubMed shortly after receipt of source files and the completed license to publish. Please notify sourcefiles@thegsajournals.org if you do not wish to publish your article via Advanced Access.
4. We invite you to submit an original color figure related to your paper for consideration as cover art. Please email your submission to the editorial office or upload it with your final files. You can submit a small-sized image for evaluation, and if selected, the final image must be a TIFF file 2513px wide by 3263px high (8.375 by 10.875 inches; resolution of 600ppi). Please avoid graphs and small type.
5. Let us know if this paper is from a recently-established lab (within the last 5 years) so it can be considered for extra visibility.

If you have any questions or encounter any problems while uploading your accepted manuscript files, please email the editorial office at sourcefiles@thegsajournals.org.

Sincerely,

Aneil Agrawal
Associate Editor
GENETICS

Approved by:
Nicholas Barton
Senior Editor
GENETICS

note: Please add jnls.author.support@oup.com and genetics.oup@novatechset.com (or the domains @oup.com and novatechset.com) to your email program's "safe senders" list. You will be contacted by both at various points during the

production process.

Review comments (if applicable):